# Continuous Repetition Motor Imagery Training and Physical Practice Training Exert the Growth of Fatigue and Its Effect on Performance

**DOI:** 10.3390/brainsci12081087

**Published:** 2022-08-16

**Authors:** Akira Nakashima, Takefumi Moriuchi, Daiki Matsuda, Jirou Nakamura, Kengo Fujiwara, Yuta Ikio, Takashi Hasegawa, Wataru Mitunaga, Toshio Higashi

**Affiliations:** 1Department of Rehabilitation, Juzenkai Hospital, Nagasaki 852-8012, Japan; 2Graduate School of Biomedical Sciences, Nagasaki University, Nagasaki 852-8012, Japan

**Keywords:** central fatigue, corticospinal excitability, motor imagery, motor performance, transcranial magnetic stimulation

## Abstract

Continuous repetition of motor imagery leads to mental fatigue. This study aimed to examine whether fatigue caused by motor imagery training affects improvement in performance and the change in corticospinal excitability. The participants were divided into “physical practice training” and “motor imagery training” groups, and a visuomotor task (set at 50% of maximal voluntary contraction in participants) was performed to assess the training effect on fatigue. The measurements were recorded before and after training. Corticospinal excitability at rest was measured by transcranial magnetic stimulation according to the Neurophysiological Index. Subjective mental fatigue and muscle fatigue were assessed by using the visual analog scale and by measuring the pinch force, respectively. Additionally, the error area was evaluated and calculated at pre-, mid-, and post-terms after training, using a visuomotor task. After training, muscle fatigue, subjective mental fatigue, and decreased corticospinal excitability were noted in both of the groups. Moreover, the visuomotor task decreased the error area by training; however, there was no difference in the error area between the mid- and post-terms. In conclusion, motor imagery training resulted in central fatigue by continuous repetition, which influenced the improvement in performance in the same manner as physical practice training.

## 1. Introduction

Fatigue can be defined as difficulty in initiating or sustaining voluntary activities [1]. Muscle fatigue is defined as an exercise-induced decrease in the maximal voluntary force or power produced by a muscle or a muscle group [2], which may develop as a result of peripheral changes in the muscle level and failure of the central nervous system to adequately drive the motoneurons [3]. The latter, called “central fatigue”, is defined as a progressive decline in the ability to voluntarily activate muscles [4]. The studies on transcranial magnetic stimulation (TMS) have demonstrated a decrease in the excitability of the primary motor cortex following muscle fatigue, using motor-evoked potentials (MEPs) [5,6]. Moreover, the corticomotor excitability shifts to the somatosensory cortex region to enable recovery from muscle fatigue [7]. In addition, previous research showed that the decrease in MEP after muscle fatigue was induced by tetanic electrical stimulation [8]. The main source of muscle fatigue is the increased inhibitory effect exerted by group III and IV afferent nerves, which carry sensory information to the central nervous system and motor neurons in the spinal cord [9]. Moreover, a reduced excitatory input from muscle spindles [10] was reported during muscle fatigue. The brain mechanisms increase the activity of the inhibitory networks mediated by the GABAB receptors [11], and the right dorsolateral prefrontal cortex is involved in the neural substrates of central inhibition during physical fatigue [12].

A previous study used electromyography (EMG) to demonstrate muscle fatigue at 3 min after a hand-grip endurance motor-imagery task [13]. Motor imagery (MI) is defined as “mental simulation” or “mental rehearsal” of movement without any actual body movement [14,15]. It has been reported that in MI, activation in the brain is similar to the actual action, without the accompanying movement. Specifically, some of the reviews revealed that there are similar brain areas that activate in both MI and actual movements, such as the premotor area, supplementary motor area, inferior parietal lobule, superior parietal lobule, cerebellum, basal ganglia, and the prefrontal cortex [16,17]. That is, muscle fatigue does not cause peripheral fatigue after motor imagery; rather, we believe that the underlying cause is central fatigue. Recently, we investigated the multi-faceted issue of the development of fatigue during continuously repeated motor imagery, and found that the repetition of motor imagery decreased the concentration and the excitability of the corticospinal tract, in addition to producing muscle fatigue and mental fatigue [18]. However, only a few reports are available on the relationship between improved performance and fatigue as a result of the continuous repetition of motor imagery [13,18,19,20]. Therefore, little is known regarding the effects of the fatigue caused by the continuous repetition of motor imagery on improvement in performance.

We divided individuals into the “physical practice training” and “motor imagery training” groups and used a visuomotor task to investigate the effects of motor imagery training on fatigue. This study principally aimed to examine whether the fatigue caused by motor imagery training affects the improvement in performance and the change in corticospinal excitability. We hypothesized that the motor imagery training, similar to physical practice training, would negatively affect performance by creating fatigue.

## 2. Materials and Methods

### 2.1. Participants

This study was based on the Global Guidelines for Care in the Use of TMS [21]. In the first stage of recruitment, all of the participants completed a questionnaire designed to exclude those with contraindications. No participants reported any neurological impairments or contraindications to TMS. In total, 24 participants (four female individuals; mean age, 25.7 ± 3.6 years) were included in this experiment. All of the participants were self-reported right-handers. They were divided into the physical practice training (*n* = 12) (PP group) and the motor imagery training groups (*n* = 12) (MI group). The motor imagery ability of the participants in the MI group was evaluated using the Movement Imagery Questionnaire-Revised (minimum score, 8 points; maximum score, 56 points) [22]. The mean motor image ability of the participants in the MI group was found to be 48.1 ± 6 (kinesthetic imagery, 25.2 ± 1.8; visual imagery, 22.6 ± 4.6). There were no problems with the motor imagery ability in the MI group participants.

The study was approved by the local Ethics Committee of the Nagasaki University Graduate School of Biomedical and Health Sciences (approval no.: 18121302-3; date of approval: 2 October 2020). All of the experimental procedures were conducted in accordance with the principles embodied in the Declaration of Helsinki.

### 2.2. EMG Recording

The surface EMG activity was recorded in the right abductor pollicis brevis (APB) muscle, using a pair of Ag-AgCl cup electrodes with a 9-mm diameter (SDC112; GE Healthcare, Japan). This study focused on the APB muscle, as it was the main agonist muscle during the task. The recording and reference electrodes were placed over the muscle and the tendon. The surface EMG signals were amplified and filtered at a bandwidth of 5–3000 Hz, using a digital signal processor (Neuropack Sigma MEB-5504; Nihon Kohden, Japan) and transferred to a computer for offline analysis after passage through an A/D converter at a sampling frequency of 2 kHz (PowerLab 16/30; AD Instruments, Bella Vista, Australia). Throughout the experiment, the participants were instructed to avoid inadvertent movements that could lead to background EMG activity. For each muscle in each trial, a 20-ms period preceding the TMS trigger was assessed for any background EMG activity, and the trials with a background EMG activity of >20 µV were excluded from the analysis.

### 2.3. TMS

We used a 70-mm figure-of-eight coil connected to a magnetic TMS stimulator (Magstim 200; Magstim, UK). At the beginning of the experiment, we identified the optimal TMS coil position for evoking MEPs in the right APB muscle (the hotspot). Then, we marked the coil position on the swimming cap covering each individual’s scalp. The coil was placed tangentially to the scalp with its handle pointing backward at an angle of approximately 45° away from the midsagittal line. Care was taken to maintain the same coil position relative to the scalp throughout the experiment. The resting motor threshold (MT) was defined as the lowest stimulus intensity that evoked an MEP ≥ 50 µV in amplitude in the right APB muscle in five out of 10 trials. The intensity of the test stimulus was set to 110–130% of the resting MT, and the mean size of the control MEP for the APB muscle was 0.75–1.25 mV. Furthermore, the trials with background EMG activity greater than 20 μV were eliminated from the analysis. The MEP amplitude (peak-to-peak) was measured from APB muscle. The MEP amplitude was analyzed using peak-to-peak values and expressed as an absolute value.

### 2.4. Measurements

Mental fatigue was evaluated using the visual analog scale (VAS). The participants were instructed to draw a line on paper of which the length represented their muscle or mental fatigue (0 mm = “I am not physically/mentally fatigued at all”; 100 mm = “I am physically/mentally very fatigued”). The muscle fatigue was assessed using pinch force. The pinch force (kg) was evaluated using a hydraulic pinch gauge (SH-5005, Sakai Medical, Tokyo, Japan). For the pinch force, the participants were asked to put into force at maximal voluntary contraction (MVC) for 2000 ms.

### 2.5. Visuomotor Task

A computerized pulse-generation system (LabView; National Instruments, Austin, TX, USA) was used to design the visuomotor task. The task result was the output of forces from the right thumb and right index finger and was set to 50% of the MVC. The prehension object was a button (30 × 40 × 10 mm), in the middle of which a strain sensor (LMB-A-200N-P; Kyowa, Tokyo, Japan) was placed (Figure 1B). First, the participants were visually assessed to ensure that the reaction marker (red circle) automatically moved from left to right on the orbit axis, which was displayed on a computer screen (Figure 1C). Then, we instructed the participants that the output of force should be 50% of the MVC for 2000 ms. The start signal was delivered 4000 ms after the warning signal. One session lasted 10s, and the error area was calculated for the performance indicator. The error area was calculated by subtracting the area of the target waveform from the area of the actual output waveform, using the LabView program. The error area was calculated for each session, and the average value was recorded for a set of 10 measurements. 

### 2.6. Physical Practice Training and Motor Imagery Training

The participants were asked to execute physical practice training and motor imagery training sitting on a chair with both hands on a table (Figure 1A). Each practice training session comprised 10 sets of 20 trial sessions involving a visuomotor task. The PP group was asked to perform output of force with 50% of the MVC with visual feedback, using the right thumb and the right index finger. The MI group was asked to grasp the button, and the instructed output force was 50% of MVC by motor imagery. During the imagery task, the participants were instructed to imagine performing this movement. 

### 2.7. Experimental Procedures

This research was conducted in two groups: the PP and the MI groups. The mental fatigue was assessed using the VAS. Additionally, regarding the pinch force, the aim was to obtain the MVC for 2000 ms. Then, a 10-trial visuomotor task set at 50% of the MVC was performed. The experiment was conducted with the participant in a comfortable posture with both of the upper limbs resting on a table. Prior to the experiment, the corticospinal excitability in the resting state was assessed by acquiring 10 MEPs, while the participant passively watched a white-colored fixation cross on a black background at the center of the computer screen. Then, the experiment comprised 10 sets of 20 trial sessions involving the visuomotor task. The visuomotor task was performed in the PP and MI groups using physical training and motor imagery, respectively. The PP group was asked to use an output of 50% MVC force, using the visual reaction markers projected on the monitor for feedback. The MI group was asked to perform motor imagery while recalling the 50% MVC force from the performance test used in the pre-evaluation. The MI group was controlled by two types of sounds as the participants carried out motor imagery, while passively watching a white-colored fixation cross on a black background at the center of the computer screen. A session started with a warning signal, and a start signal was delivered 4000 ms later. The task was the output of forces from the right thumb and right index finger for a precise grasp of the prehension object. The instructed output force was 50% of the MVC by motor imagery for 2000 ms. During the motor imagery task, the participants were instructed to imagine performing this movement. Specifically, they were asked to imagine the kinesthetic experience of the movement (rather than a visual type of imagery), remain relaxed, and avoid movements during the task. An interim appraisal assessed the visuomotor task after 100 training sessions. The post-evaluations were conducted in a similar manner to the pre-evaluations (Figure 2).

### 2.8. Statistical Analysis

First, all of the variables were tested for normality using the Shapiro–Wilk test by employing the difference in the outcome before and after each training session. Only the pinch force evaluation indicators did not satisfy the normal distribution. As a result, an analysis of variance (ANOVA) was used for the normally distributed outcome values with the time point as the within-subject factor and the training program as the between-subject factor. Mauchly’s test was used for the error area data, to assess the statistical assumption of sphericity. The post-hoc analysis, using the Bonferroni method, was employed for the comparisons of error areas for the temporal changes due to the training. The non-normally distributed outcome values (pinch force) were analyzed using the Wilcoxon signed-rank test. Therefore, to investigate the degree of muscle weakness associated with the training, we compared the pinch force values before and after the intervention in each group. In all of the analyses, the threshold for statistical significance was set to *p* < 0.05. All of the analyses were performed using a statistical analysis software (SPSS version 22.0; IBM Corp., Armonk, NY, USA).

## 3. Results

Only the pinch force did not have a normal distribution. The ANOVA revealed a main effect of “time” on the VAS (mental fatigue) (F_(1,22)_ = 77.738; *p* < 0.001) (Figure 3), the MEP (F_(1,22)_ = 21.066; *p* < 0.001) (Figure 4), and the error area (F_(2,44)_ = 26.409; *p* < 0.001). The ANOVA also revealed a main effect of the “training program” on the VAS (mental fatigue) (F_(1,22)_ = 8.120; *p* = 0.009) and the error area (F_(1,22)_ = 9.051; *p* = 0.006); however, the MEP was not found to have a main effect (F_(1,22)_ = 0.879; *p* = 0.359). There was no significant interaction between “time × training program” on the VAS (mental fatigue) (F_(1,22)_ = 1.367; *p* = 0.255), MEP (F_(1,22)_ = 0.725; *p* = 0.404), and the error area (F_(2,44)_ = 1.862; *p* = 0.167). Moreover, Mauchly’s test applied to the error area data showed a *p*-value of 0.389. The post-hoc analysis using the Bonferroni method revealed a significant decrease in the error area by midterm (*p* < 0.001) and post-term (*p* < 0.001), compared with that by the pre-term in all of the participants. However, there was no significant differences between the midterm and post-term (*p* = 1.000) (Figure 5). The Wilcoxon signed-rank test revealed a significant decrease in the pinch force after training in the PP (*p* = 0.002) and MI (*p* = 0.002) groups (Figure 6).

## 4. Discussion

This study aimed to examine whether the fatigue caused by the motor imagery training affected the improvement in performance and corticospinal excitability. This research was conducted using a visuomotor task with 50% MVC, after dividing participants into the PP and MI groups. We found that muscle fatigue, subjective mental fatigue, and decreased corticospinal excitability occurred in both of the groups after training. Moreover, the visuomotor task significantly decreased the error area after training between 0 and 100 times. However, the error area did not significantly change after training between 100 and 200 times. The study revealed that both the physical practice training and the motor imagery training caused fatigue, which in turn affected the performance.

The PP group showed decreased muscle strength and corticospinal excitability after training. Our results are consistent with those of published studies [5,6,23]. In addition, the visuomotor task could fully trigger fatigue, as it induced both muscle fatigue and subjective mental fatigue. Moreover, the muscle strength and corticospinal excitability decreased even in the MI group following training. Motor imagery is defined as “mental simulation” or “mental rehearsal” of movement, without any actual body movement [14,15]. Therefore, we hypothesized that the central fatigue decreases the muscle strength and corticospinal excitability. Thus, decreased muscle strength was considered to reflect poor motor-nerve activation, which explains why the motor imagery training resulted in central fatigue. The neurophysiological TMS studies revealed a decrease in corticospinal excitability related to the contraction of the target muscle using MEP after the motor imagery training with a handgrip endurance task [24]. Furthermore, a previous electroencephalography study reported an increase in fatigue strength during motor imagery, which increased in the alpha, delta, and theta bands [25]. However, further investigation is needed to understand the underlying relationship between motor imagery and decreased muscle strength.

There was a significant difference between the pre- and mid-terms as well as between the pre- and post-terms after training in both of the groups, demonstrating that a continuous repetition of motor imagery can improve the performance of movement tasks [26,27]. In fact, combining mental training and physical training is effective [19]; it has been reported that the capacity to form vivid “movement images” can enhance the effects of training [28]. In addition, many elite athletes state that they use motor imagery to improve performance. Moreover, many studies have confirmed that the motor imagery practice can be useful for improving performance in rehabilitation programs [29]. A previous study reported that the motor imagery training led to a significant improvement in performance; however, the improvement was significantly less than that produced by physical practice training [26]. Furthermore, neither the PP nor MI group showed a significant improvement in performance between the mid- and post-terms after training. This could be explained by the fact that the MEP significantly decreased after training in both of the groups. While the PP group experienced increased muscle and mental fatigue due to peripheral and central fatigue, the MI group experienced muscle and mental fatigue due to central fatigue alone. Mental fatigue is a sensation that develops after prolonged cognitive activity [30] and affects performance [31]. A previous study reported that learning in a fatigued state results in detrimental effects on the overall task acquisition [32]. In the present study, we found an increase in central fatigue by the continuous repetition of motor imagery. This can have an impact on improved performance. The peripheral fatigue did not occur in the motor imagery. Altogether, from the perspective of the proportion of fatigue, we speculated that the motor imagery training caused stronger central fatigue in comparison to physical practice.

This study had several limitations, including its sample size. Twelve participants were included in each group; however, the number of participants was insufficient to reveal an improvement in performance and fatigue by training. In addition, we did not measure central fatigue using a more appropriate technique, such as the twitch interpolation technique and normalizing MEPs to an M-wave evoked with peripheral nerve stimulation. Therefore, we need to continue our study in this area.

## 5. Conclusions

In summary, the motor imagery training resulted in mental fatigue by the continuous repetition of motor imagery, which influenced the improvement in the performance in the same manner as physical practice training. Our experimental design motor imagery training did not enhance performance. Therefore, it is necessary to set a motor imagery training protocol, considering the fatigue that it causes. In addition, mental fatigue affects the central nervous system and is thought to affect muscle strength, which decreases corticospinal excitability [18]. The results of this study may aid in establishing a motor imagery protocol that considers fatigue. Thus, careful consideration is required when designing training protocols in the field of rehabilitation and sports, as excess fatigue associated with motor imagery training may have a negative effect on improved performance.

## Figures and Tables

**Figure 1 brainsci-12-01087-f001:**
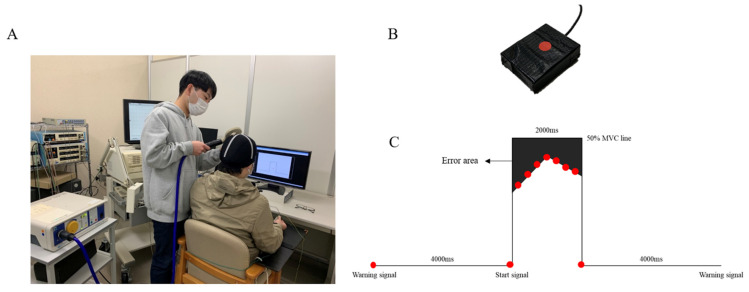
Experimental apparatus. (**A**) Experimental setup; (**B**) The strain sensor is placed in the middle of the prehension object; (**C**) Visuomotor Task; the tracking index is displayed on the computer screen in front of the participants. The red circle indicates the reaction marker.

**Figure 2 brainsci-12-01087-f002:**
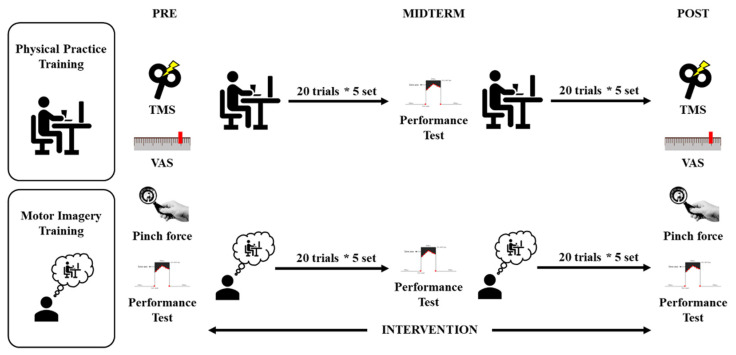
Experimental protocol. TMS, transcranial magnetic stimulation; VAS, visual analog scale; *, multiply.

**Figure 3 brainsci-12-01087-f003:**
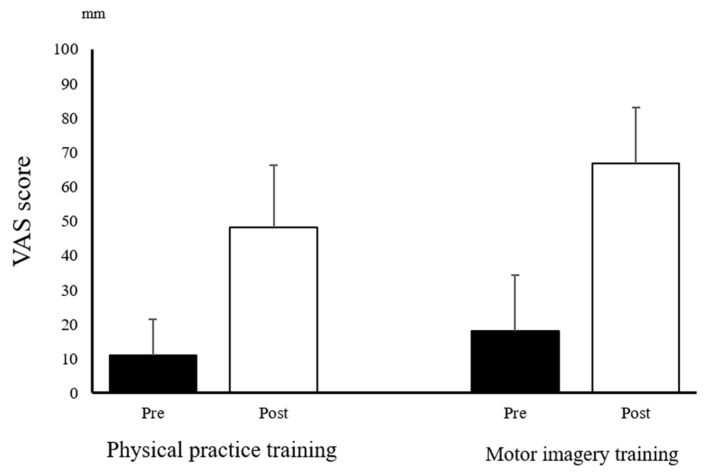
VAS before and after training. VAS, visual analog scale.

**Figure 4 brainsci-12-01087-f004:**
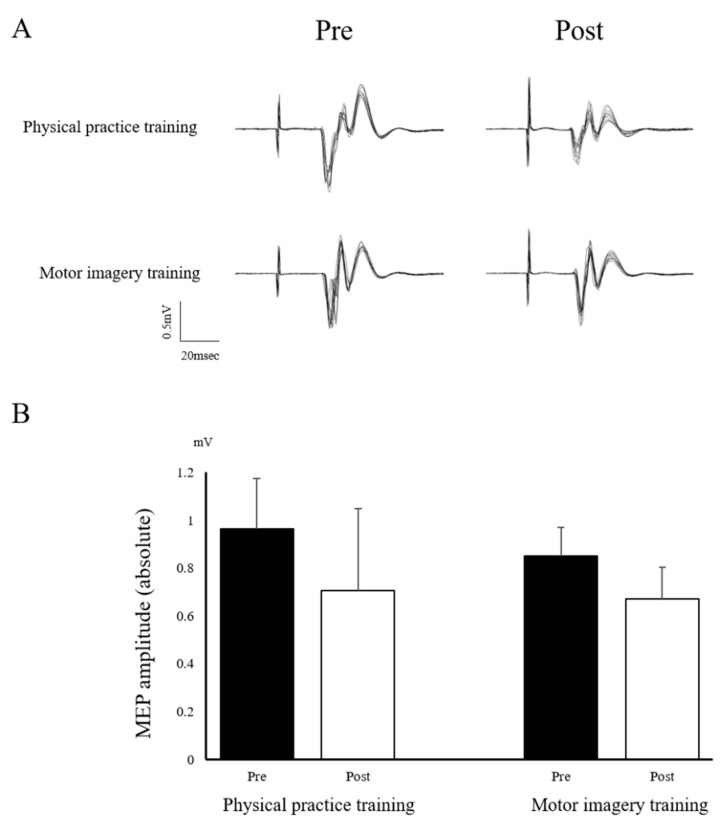
Neurophysiological findings before or after training. (**A**) Typical superimposed waveforms of MEP amplitudes in the right APB muscle obtained in 10 trials before and after training in the two groups; (**B**) The absolute MEP amplitude before and after training. MEP, motor-evoked potential; APB, abductor pollicis brevis.

**Figure 5 brainsci-12-01087-f005:**
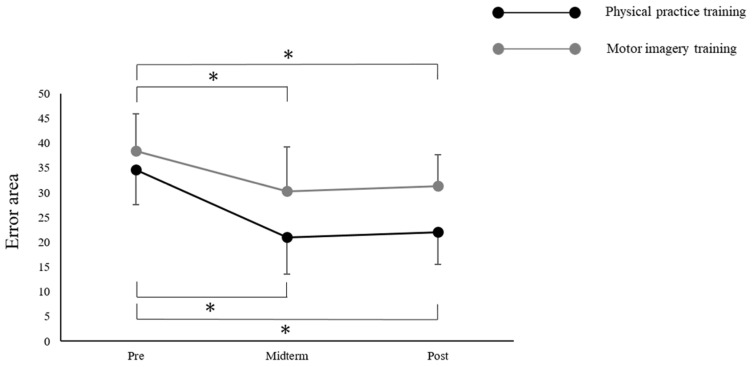
Chronological changes in the error area during training. * Significant difference between pre- and mid-terms as well as pre- and post-terms, *p* < 0.05. Data are expressed as means ± standard errors.

**Figure 6 brainsci-12-01087-f006:**
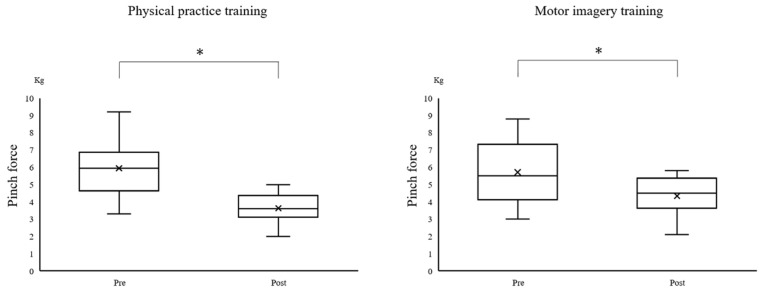
Comparison of the pinch force values before and after the intervention in each group. * Significant difference between pre- and post-terms, *p* < 0.05.

## Data Availability

The datasets generated and/or analyzed during the current study are available from the corresponding author on reasonable request.

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
