# Peer review of "Continuous Repetition Motor Imagery Training and Physical Practice Training Exert the Growth of Fatigue and Its Effect on Performance"

_brainsci, 2022, doi:10.3390/brainsci12081087_

Round 1

Reviewer 1 Report

Dear Editor,

Thank you very much for giving me the opportunity to review this article which compared the effect of a continuous repetition of motor emagery training versus physical practice training on central fatigue. Both of the training programs decreased significantly the corticospinal excitability, which lead to the central fatigue presence for the motor emagery group whereas the physical practice group experienced both of the central and peripheral fatigue. 

Although the study has some limitations in terms of the number of participants, the normalization of MEPS (to an M-wave evoked with peripheral nerve stimulation) and the techniques used for the assessment of fatigue, the results provided are consistent with those of previous studies and could bring new recommendations in favor of mental imagery training. Not being characterized by the installation of peripheral fatigue, motor imagery training could be a means of limiting the risk of injury, especially during overload cycles.

I would like the authors to develop this aspect in the discussion by relying on references.

Author Response

AUTHORS’ RESPONSES TO REVIEWER 1’S COMMENTS

The authors would like to thank the reviewer for his/her constructive critique to improve the manuscript. We have made every effort to address the issues raised and to respond to all comments. The revisions are indicated in red font in the revised manuscript. Please find below a detailed, point-by-point response to the reviewer's comments. We hope that our revisions meet the reviewer’s expectations.

Although the study has some limitations in terms of the number of participants, the normalization of MEPS (to an M-wave evoked with peripheral nerve stimulation) and the techniques used for the assessment of fatigue, the results provided are consistent with those of previous studies and could bring new recommendations in favor of mental imagery training. Not being characterized by the installation of peripheral fatigue, motor imagery training could be a means of limiting the risk of injury, especially during overload cycles.

I would like the authors to develop this aspect in the discussion by relying on references.

Response: We appreciate the reviewer’s comment on this point. In accordance with the reviewer’s comment, we have added text to the Discussion section as follows (Line 248-253)

“There was a significant difference between pre- and mid-terms as well as between pre- and post-terms after training in both groups, demonstrating that continuous repetition of motor imagery can improve the performance of movement tasks [26,27]. In fact, combining mental training and physical training is effective [19]; it has been reported that the capacity to form vivid “movement images” can enhance the effects of training [28]. In addition, many elite athletes state that they use motor imagery to improve performance. Moreover, many studies have confirmed that motor imagery practice can be useful for improving performance in rehabilitation programs [29]. ”

Reviewer 2 Report

Thank you for permitting me to review this manuscript 

In this study authors  assessed whether fatigue caused by motor imagery training affects the improvement in performance and the change in corti cospinal excitability. 24 individuals were enrolled in 2 groups of physical practice and motor imagery both group had  fatigue and had impaired performance 

 Line 49 Please elaborate  motor imagery and describe some example if possible 

Line 52 Please provide reference  PPR

Pleas eprovide better resolution for the subtable in figure 1

Fig 2 if there was a significant differene in VAS please add an asterix  between pre and post 

Line  257  citing cardio respiratory parameters are purely speculative as in this study cradiorespiratory parameters were not assesseed , please delete or please insert additional explanations 

It is better to conclude within our experimental design motor imagery could not enhance performance 

Author Response

AUTHORS’ RESPONSES TO REVIEWER 2’S COMMENTS

The authors would like to thank the reviewer for his/her constructive critique to improve the manuscript. We have made every effort to address the issues raised and to respond to all comments. The revisions are indicated in red font in the revised manuscript. Please find below a detailed, point-by-point response to the reviewer's comments. We hope that our revisions meet the reviewer’s expectations. 

Line 49 Please elaborate motor imagery and describe some example if possible

Response: We appreciate the reviewer’s comment on this point. In accordance with the reviewer’s comment, we have added text to the Introduction section as follows (Lines 49-55):

“Motor imagery (MI) is defined as “mental simulation” or “mental rehearsal” of movement without any actual body movement [14,15]. It has been reported that in MI, activation in the brain is similar to the actual action, without the accompanying movement. Specifically, some reviews revealed that there are similar brain areas that activate in both MI and actual movements, such as the premotor area, supplementary motor area, inferior parietal lobule, superior parietal lobule, cerebellum, basal ganglia, and the prefrontal cortex [16,17]. ”

Line 52 Please provide reference PPR

Response: We appreciate the reviewer’s comment on this point. In accordance with the reviewer’s comment, we have changed the text in the Introduction section as follows (Line 57-60):

<original>” Recently, we demonstrated that repetition of motor imagery decreases the excitability of the corticospinal tract in addition to producing muscle fatigue and mental fatigue”

<revised>“Recently, we investigated the multi-faceted issue of the development of fatigue during continuously repeated motor imagery, and found that repetition of motor imagery decreased concentration and the excitability of the corticospinal tract in addition to producing muscle fatigue and mental fatigue.”

Pleas eprovide better resolution for the subtable in figure 1

Response: We appreciate the reviewer’s comment on this point. In accordance with the reviewer’s comment, we have changed Figure 1.

Fig 2 if there was a significant differene in VAS please add an asterix between pre and post

Response: We appreciate the reviewer’s comment on this point. We used normally distributed outcome values with the time point as the within-subject factor, and the training program as the between-subject factor, analyzed by ANOVA. Among them, the VAS score and MEP showed a main effect on time point by ANOVA. We do not consider these two assessments to be amenable to a post-hoc test because they were measured before and after the training session. For this reason, we don't think it is necessary to add an asterisk.

Line  257  citing cardio respiratory parameters are purely speculative as in this study cradiorespiratory parameters were not assesseed , please delete or please insert additional explanations

Response: We appreciate the reviewer’s comment on this point. In accordance with the reviewer’s comment, we have deleted the text in the Discussion session.

< original >” A previous study in physical practice confirmed experimentally the hypothesis that mental fatigue impairs physical performance in humans. The results of this study provided experimental evidence that mental fatigue limits exercise tolerance in humans through higher perception of effort rather than cardiorespiratory and musculoenergetic mechanisms [30].”

<revised>None

It is better to conclude within our experimental design motor imagery could not enhance performance

Response: We appreciate the reviewer’s comment on this point. In accordance with the reviewer’s comment, we have added text to the Conclusion session. (Line 278-279)

“In summary, motor imagery training resulted in mental fatigue by continuous repetition of motor imagery, which influenced the improvement in performance in the same manner as physical practice training. Our experimental design motor imagery training did not enhance performance. Therefore, it is necessary to set a motor imagery training protocol, considering the fatigue that it causes. In addition, mental fatigue affects the central nervous system and is thought to affect muscle strength, which decreases corticospinal excitability. The results of this study may aid in establishing a motor imagery protocol that considers fatigue. Thus, careful consideration is required when designing training protocols in the field of rehabilitation and sports, as excess fatigue associated with motor imagery training may have a negative effect on improved performance.”

Reviewer 3 Report

The authors of the article " Continuous Repetition Motor Imagery Training and Physical Practice Training Exert the Growth of Fatigue and Its Effect on Performancedesigned an experimental protocol to test the hypothesis that, like physical practice training, motor imagery training would have a negative impact on performance by inducing fatigue. 

The idea of the study is intriguing, yet the manuscript raises just a few issues: 

·      When reading the protocol section, it is not quite obvious how TMS has been employed; further information is required.

·    The authors use TMS in section 2.3 to elicit a 5/10 ABP thumb contraction by probing the motor region using the Rossini et al procedure. Here, the motor region (e.x M1), not the APB muscle, is the hotspot (the hotspot). Another query: it could be helpful to comprehend the rationale for choosing a threshold of 110-130 %

·      The authors claim that MEPs were recording on the right abductor in section 2.3. The right thumb and index finger were shown to be used in a button tap in section 2.5. The timing of the recording and button hit is unclear. Additionally, the authors fail to indicate the delivery time of TMS in the manuscript. I would suggest better addressing this section by designing a great paradigm chronology of motor execution, motor imagery, and TMS application (Like what we found in BCI papers).  

·      When authors assert “Then, we instructed the participants that the output of force should be 50% of the MVC for 2000 ms. The start signal was delivered 4000 ms after the warning signal.”, How do participants compute 50 % MVC?

·      Lines 136-138 overlap 156-158. I would suggest refocusing  

·      The authors mention in section 2.8 that "First, all variables were tested for normality," however the reader is still in the dark as to the types of signals, preprocessing, feature extraction, etc. An accurate description of these steps would help readers better comprehend the theoretical underpinnings of this study.

·      Although it is not entirely evident from the research, pinch force is likely a time series; nevertheless, preprocessing and feature extraction details are lacking.

·      Based on the statistical analysis section, only "pinch force" has been statistically analyzed, therefore I'm working hard to keep up with the findings of the MEPS test and outcomes.

·      In figure 3, the first pick, in my opinion, is connected to the TMS artifact in panel A. We seldom have the ability to evaluate the MEP when TMS artifact is present, therefore this figure is odd. Additionally, the time scale (x-axe) doesn't seem correct. Probably, there is a problem with MEP processing. I'd advise preprocessing to get rid of TMS artifacts.

·      I would recommend referencing 275-277.

Author Response

AUTHORS’ RESPONSES TO REVIEWER 3’S COMMENTS

The authors would like to thank the reviewer for his/her constructive critique to improve the manuscript. We have made every effort to address the issues raised and to respond to all comments. The revisions are indicated in red font in the revised manuscript. Please find below a detailed, point-by-point response to the reviewer's comments. We hope that our revisions meet the reviewer’s expectations.

When reading the protocol section, it is not quite obvious how TMS has been employed; further information is required

Response: We appreciate the reviewer’s comment on this point. In accordance with the reviewer’s comment, we have added text to the 2.3 TMS section as follows (Line 113-117)

< original >” We used a 70-mm figure-of-eight coil connected to a magnetic TMS stimulator (Mag-stim 200, Magstim, UK). At the beginning of the experiment, we identified the optimal TMS coil position for evoking MEPs in the right APB muscle (the hotspot). Then, we marked the coil position on the swimming cap covering each individual’s scalp. The coil was placed tangentially to the scalp with its handle pointing backward at an angle of approximately 45° away from the midsagittal line. Care was taken to maintain the same coil position relative to the scalp throughout the experiment. The resting motor threshold (MT) was defined as the lowest stimulus intensity that evoked an MEP ≥50 µV in amplitude in the right APB muscle in five out of 10 trials. The intensity of the test stimulus was set to 110–130% of the resting MT, and the mean size of the control MEP for the APB muscle was 0.75–1.25 mV.”

<revised>” We used a 70-mm figure-of-eight coil connected to a magnetic TMS stimulator (Mag-stim 200, Magstim, UK). At the beginning of the experiment, we identified the optimal TMS coil position for evoking MEPs in the right APB muscle (the hotspot). Then, we marked the coil position on the swimming cap covering each individual’s scalp. The coil was placed tangentially to the scalp with its handle pointing backward at an angle of approximately 45° away from the midsagittal line. Care was taken to maintain the same coil position relative to the scalp throughout the experiment. The resting motor threshold (MT) was defined as the lowest stimulus intensity that evoked an MEP ≥50 µV in amplitude in the right APB muscle in five out of 10 trials. The intensity of the test stimulus was set to 110–130% of the resting MT, and the mean size of the control MEP for the APB muscle was 0.75–1.25 mV. Furthermore, trials with background EMG activity greater than 20 μV were eliminated from the analysis. MEP amplitude (peak-to-peak) was measured from APB muscle. MEP amplitude was analyzed using peak-to-peak values and expressed as an absolute value.”

In addition, we feel that the representation of the experimental protocol was unclear. Thus, we needed to undertake a radical overhaul of Figure 1. We separated Figure 1 into Figure 1 and Figure 2 to clarify the experimental protocol.

The authors use TMS in section 2.3 to elicit a 5/10 ABP thumb contraction by probing the motor region using the Rossini et al procedure. Here, the motor region (e.x M1), not the APB muscle, is the hotspot (the hotspot). Another query: it could be helpful to comprehend the rationale for choosing a threshold

Response : We would like to thank the reviewer for the comment. The resting motor threshold (MT) was defined as the lowest stimulus intensity that evoked an MEP ≥50 µV in amplitude in the right APB muscle in five out of 10 trials as described by the manuscript.

The authors claim that MEPs were recording on the right abductor in section 2.3. The right thumb and index finger were shown to be used in a button tap in section 2.5. The timing of the recording and button hit is unclear. Additionally, the authors fail to indicate the delivery time of TMS in the manuscript. I would suggest better addressing this section by designing a great paradigm chronology of motor execution, motor imagery, and TMS application (Like what we found in BCI papers)

Response: We appreciate the reviewer’s comment on this point. TMS was evaluated at rest in before and after training. Therefore, there was no need to match the timing of the button and the MEP recording. To clarify this point, we have added text to the 2.3 TMS section as follows (Line 113-117)

< original >” We used a 70-mm figure-of-eight coil connected to a magnetic TMS stimulator (Mag-stim 200, Magstim, UK). At the beginning of the experiment, we identified the optimal TMS coil position for evoking MEPs in the right APB muscle (the hotspot). Then, we marked the coil position on the swimming cap covering each individual’s scalp. The coil was placed tangentially to the scalp with its handle pointing backward at an angle of approximately 45° away from the midsagittal line. Care was taken to maintain the same coil position relative to the scalp throughout the experiment. The resting motor threshold (MT) was defined as the lowest stimulus intensity that evoked an MEP ≥50 µV in amplitude in the right APB muscle in five out of 10 trials. The intensity of the test stimulus was set to 110–130% of the resting MT, and the mean size of the control MEP for the APB muscle was 0.75–1.25 mV.”

<revised>” We used a 70-mm figure-of-eight coil connected to a magnetic TMS stimulator (Mag-stim 200, Magstim, UK). At the beginning of the experiment, we identified the optimal TMS coil position for evoking MEPs in the right APB muscle (the hotspot). Then, we marked the coil position on the swimming cap covering each individual’s scalp. The coil was placed tangentially to the scalp with its handle pointing backward at an angle of approximately 45° away from the midsagittal line. Care was taken to maintain the same coil position relative to the scalp throughout the experiment. The resting motor threshold (MT) was defined as the lowest stimulus intensity that evoked an MEP ≥50 µV in amplitude in the right APB muscle in five out of 10 trials. The intensity of the test stimulus was set to 110–130% of the resting MT, and the mean size of the control MEP for the APB muscle was 0.75–1.25 mV. Furthermore, trials with background EMG activity greater than 20 μV were eliminated from the analysis. MEP amplitude (peak-to-peak) was measured from APB muscle. MEP amplitude was analyzed using peak-to-peak values and expressed as an absolute value.”

In addition, we feel that the representation of the experimental protocol was unclear. Thus, we needed to undertake a radical overhaul of Figure 1. We separated Figure 1 into Figure 1 and Figure 2 to clarify the experimental protocol.

When authors assert “Then, we instructed the participants that the output of force should be 50% of the MVC for 2000 ms. The start signal was delivered 4000 ms after the warning signal.”, How do participants compute 50 % MVC?

Response : We would like to thank the reviewer for the comment. The PP group was asked to use an output of 50% MVC force, using the visual reaction markers projected on the monitor for feedback. The MI group was asked to perform motor imagery while recalling the 50% MVC force from the performance test used in pre-evaluation. To clarify this point, we have added text to the 2.7. Experimental Procedures section as follows (Line 159-162):

< original >This research was conducted in two groups: the PP and MI groups. Mental fatigue was assessed using the VAS. Additionally, regarding pinch force, the aim was to obtain the MVC for 2000 ms. Then, a 10-trial visuomotor task set at 50% of the MVC was performed. The experiment was conducted with the participant in a comfortable posture with both upper limbs resting on a table. Prior to the experiment, the corticospinal excitability in the resting state was assessed by acquiring 10 MEPs, while the participant passively watched a white-colored fixation cross on a black background at the center of the computer screen. Then, the experiment comprised 10 sets of 20 trial sessions involving the visuomotor task. The visuomotor task was performed in the PP and MI groups using physical training and motor imagery, respectively. The motor imagery task was controlled by two types of sounds as the participants carried out motor imagery, while passively watching a white-colored fixation cross on a black background at the center of the computer screen. A session started with a warning signal, and a start signal was delivered 4000 ms later. The task was the output of forces from the right thumb and right index finger for a precise grasp of the prehension object. The instructed output force was 50% of the MVC by motor imagery for 2000 ms. During the motor imagery task, the participants were instructed to imagine performing this movement. Specifically, they were asked to imagine the kinesthetic experience of the movement (rather than a visual type of imagery), remain relaxed, and avoid movements during the task. An interim appraisal assessed the visuomotor task after 100 training sessions. Post-evaluations were conducted similar to pre-evaluations (Figure 2).

<revised>This research was conducted in two groups: the PP and MI groups. Mental fatigue was assessed using the VAS. Additionally, regarding pinch force, the aim was to obtain the MVC for 2000 ms. Then, a 10-trial visuomotor task set at 50% of the MVC was per-formed. The experiment was conducted with the participant in a comfortable posture with both upper limbs resting on a table. Prior to the experiment, the corticospinal excitability in the resting state was assessed by acquiring 10 MEPs, while the participant passively watched a white-colored fixation cross on a black background at the center of the computer screen. Then, the experiment comprised 10 sets of 20 trial sessions involving the visuomotor task. The visuomotor task was performed in the PP and MI groups using physical training and motor imagery, respectively. The PP group was asked to use an output of 50% MVC force, using the visual reaction markers projected on the monitor for feedback. The MI group was asked to perform motor imagery while recalling the 50% MVC force from the performance test used in pre-evaluation. The MI group was controlled by two types of sounds as the participants carried out motor imagery, while passively watching a white-colored fixation cross on a black background at the center of the computer screen. A session started with a warning signal, and a start signal was delivered 4000 ms later. The task was the output of forces from the right thumb and right index finger for a precise grasp of the prehension object. The instructed output force was 50% of the MVC by motor imagery for 2000 ms. During the motor imagery task, the participants were instructed to imagine performing this movement. Specifically, they were asked to imagine the kinesthetic experience of the movement (rather than a visual type of imagery), remain relaxed, and avoid movements during the task. An interim appraisal assessed the visuomotor task after 100 training sessions. Post-evaluations were conducted similar to pre-evaluations (Figure 2).

Lines 136-138 overlap 156-158. I would suggest refocusing

Response: We appreciate the reviewer’s comment on this point. In accordance with the reviewer’s comment, we have deleted text in the 2.6. Physical Practice Training and Motor Imagery Training session.

< original >” Specifically, they were asked to imagine the (kinesthetic) experience of the movement (rather than a visual type of imagery), remain relaxed, and avoid making movements during the imagery task.”

<revised>” Specifically, they were asked to imagine the (kinesthetic) experience of the movement (rather than a visual type of imagery), remain relaxed, and avoid making movements during the imagery task.”

The authors mention in section 2.8 that "First, all variables were tested for normality," however the reader is still in the dark as to the types of signals, preprocessing, feature extraction, etc. An accurate description of these steps would help readers better comprehend the theoretical underpinnings of this study.

Response: We appreciate the reviewer’s comment on this point. In accordance with the reviewer’s comment, we have added text to the 2.3 TMS section as follows (Line 113-117)

< original >” We used a 70-mm figure-of-eight coil connected to a magnetic TMS stimulator (Mag-stim 200, Magstim, UK). At the beginning of the experiment, we identified the optimal TMS coil position for evoking MEPs in the right APB muscle (the hotspot). Then, we marked the coil position on the swimming cap covering each individual’s scalp. The coil was placed tangentially to the scalp with its handle pointing backward at an angle of approximately 45° away from the midsagittal line. Care was taken to maintain the same coil position relative to the scalp throughout the experiment. The resting motor threshold (MT) was defined as the lowest stimulus intensity that evoked an MEP ≥50 µV in amplitude in the right APB muscle in five out of 10 trials. The intensity of the test stimulus was set to 110–130% of the resting MT, and the mean size of the control MEP for the APB muscle was 0.75–1.25 mV.”

<revised>” We used a 70-mm figure-of-eight coil connected to a magnetic TMS stimulator (Mag-stim 200, Magstim, UK). At the beginning of the experiment, we identified the optimal TMS coil position for evoking MEPs in the right APB muscle (the hotspot). Then, we marked the coil position on the swimming cap covering each individual’s scalp. The coil was placed tangentially to the scalp with its handle pointing backward at an angle of approximately 45° away from the midsagittal line. Care was taken to maintain the same coil position relative to the scalp throughout the experiment. The resting motor threshold (MT) was defined as the lowest stimulus intensity that evoked an MEP ≥50 µV in amplitude in the right APB muscle in five out of 10 trials. The intensity of the test stimulus was set to 110–130% of the resting MT, and the mean size of the control MEP for the APB muscle was 0.75–1.25 mV. Furthermore, trials with background EMG activity greater than 20 μV were eliminated from the analysis. MEP amplitude (peak-to-peak) was measured from APB muscle. MEP amplitude was analyzed using peak-to-peak values and expressed as an absolute value.”

Although it is not entirely evident from the research, pinch force is likely a time series; nevertheless, preprocessing and feature extraction details are lacking.

Response: We appreciate the reviewer’s comment on this point. Pinch force (kg) was evaluated before and after training using a hydraulic pinch gauge (SH-5005, Sakai Medical, Japan). In addition, we feel that the representation of the experimental protocol was unclear. Thus, we needed to undertake a radical overhaul of Figure 1. We separated Figure 1 into Figure 1 and Figure 2 to clarify the experimental protocol.

Based on the statistical analysis section, only "pinch force" has been statistically analyzed, therefore I'm working hard to keep up with the findings of the MEPS test and outcomes.

Response: We appreciate the reviewer’s comment on this point. We analyzed not only pinch force but also other factors in this study, as follows:

“First, all variables were tested for normality using the Shapiro–Wilk test by using the difference in outcome before and after each training. Only pinch force evaluation indicators did not satisfy the normal distribution. As a result, an analysis of variance (ANOVA) was used for normally distributed outcome values with the time point as the with-in-subject factor and the training program as the between-subject factor. Mauchly’s test was used for the error area data to assess the statistical assumption of sphericity. Post hoc analysis using the Bonferroni method was used for comparisons of error areas for temporal changes due to training. Non-normally distributed outcome values (pinch force) were analyzed using the Wilcoxon signed-rank test. Therefore, to investigate the degree of muscle weakness associated with the training, we compared the pinch force values before and after the intervention in each group. In all analyses, the threshold for statistical significance was set to p < 0.05. All analyses were performed using a statistical analysis software (SPSS version 22.0, IBM Corp., Armonk, NY, USA). “

In figure 3, the first pick, in my opinion, is connected to the TMS artifact in panel A. We seldom have the ability to evaluate the MEP when TMS artifact is present, therefore this figure is odd. Additionally, the time scale (x-axe) doesn't seem correct. Probably, there is a problem with MEP processing. I'd advise preprocessing to get rid of TMS artifacts.

Response: We appreciate the reviewer’s comment on this point. As you pointed out, the first pick is TMS artifact. However, MEP amplitude was measured from APB muscle after 20msec~30msecs from the trigger point. For this reason, we think that the MEP measure was not a concern.

I would recommend referencing 275-277.

Response: We appreciate the reviewer’s comment on this point. In accordance with the reviewer’s comment, we have added references to the Conclusion section. (Line 282)

Round 2

Reviewer 3 Report

The authors performed a good job of addressing criticisms, and the article now sounds better.